# FED-COR: FEDERATED CORRELATION TEST WITH SECURE AGGREGATION

## ABSTRACT

In this paper, we propose the first federated correlation test framework compatible with secure aggregation, namely FED-COR. In FED-COR, correlation tests are recast as frequency moment estimation problems. To estimate the frequency moments, the clients collaboratively generate a shared projection matrix and then use stable projection to encode the local information in a compact vector. As such encodings can be linearly aggregated, secure aggregation can be applied to conceal the individual updates. We formally establish the security guarantee of FED-COR by proving that only the minimum necessary information (*i.e.*, the correlation statistics) is revealed to the server. The evaluation results show that FED-COR achieves good accuracy with small client-side computation overhead and performs comparably to the centralized correlation test in several real-world case studies.

## 1  INTRODUCTION

Correlation test, as the name implies, is the process of examining the correlation between two random variables using observational data. It is a fundamental building block in a wide variety of real-world applications, including feature selection (Zheng et al., 2004), cryptanalysis (Nyberg, 2001), causal graph discovery (Spirtes et al., 2000), empirical finance (Ledoit & Wolf, 2008; Kim & Ji, 2015), medical studies (Kassirer, 1983) and genomics (Wilson et al., 1999; Dudoit et al., 2003). Because the observational data used in the correlation tests may contain sensitive information such as genomic information, and collecting participants' information to a central repository poses a significant privacy risk. To address this problem, we utilize the federated setting, where each client maintains its own data and communicates with a central server to calculate a function. The communication transcript should contain as little information as feasible to prevent the server from inferring sensitive information.

To motivate our work and ease the understanding of the problem setting, we consider a medical company that wants to study the correlation between genetic defects and races using the patients' private data from several hospitals. For a traditional method in the federated setting, the server, which is the medical company, will aggregate the hospitals' local private contingency tables[1] using secure aggregation (Bonawitz et al., 2017; Bell et al., 2020). The company can conduct correlation tests with the aggregated global contingency table without directly accessing the individual hospitals' private data. Attentive readers might be aware that the method mentioned above leaks the joint distribution, which is the whole global contingency table, to the server. The joint distribution may contain sensitive information, and leaking it will probably violate privacy regulations. For instance, the medical company can observe the genetic distribution across races from the global table.

The secure aggregation primarily supports linear aggregation. However, in correlation tests, the computation involves computing a summed $p$-th moment over the aggregated data, where $p \in (0, 1) \cup (1, 2]$. Thus, the joint distribution will be leaked if we directly apply secure aggregation. To bridge the gap between secure aggregation and federated correlation tests, we take an important step towards designing non-linear secure aggregation protocols. Specifically, we design a federated protocol framework, namely FED-COR, optimized for a class of correlation tests, such as $\chi^2$-test and G-test. FED-COR is designed to *have low computation and communication costs* and *only disclose information that is much less sensitive than the joint distribution*. Our first insight is to recast correlation tests as frequency moment estimation problems. To approximate the frequency moments

---

[1]Contingency table contains the frequency distribution of the variables; see (Wikipedia, 2021).

in a federated manner, each client collaborates with the other clients to generate a projection matrix and encodes its raw data into a low-dimensional vector via stable random projection (Indyk, 2006; Vempala, 2005; Li, 2008). Such encodings can be aggregated with only summation, allowing clients to leverage secure aggregation to aggregate the encodings. The server then decodes the aggregated encoding to approximate the frequency moments. As secure aggregation conceals each client's individual update within the aggregated global update, the server learns only necessary information for the correlation test.

To illustrate the power of FED-COR, we instantiate it with a representative correlation test, namely Pearson's $\chi^2$-test (Pearson, 1900) and refer to the concrete protocol as FED-$\chi^2$. We evaluate FED-$\chi^2$ on 4 synthetic datasets and 16 real-world datasets. The results show that FED-$\chi^2$ can replace centralized correlation tests with good accuracy. Compared to the traditional method with secure aggregation mentioned above, FED-$\chi^2$ saves a factor of $\mathcal{O}(m)$ communication cost per client, where $m$ is the size of the contingency tables. In FED-$\chi^2$, clients only need to upload a low-dimensional encoding with size $\ell \ll m$, while in the traditional method the clients will upload the complete contingency tables. Additionally, we analyze FED-$\chi^2$ in two real-world use cases: feature selection and online false discovery rate control. The results show that FED-$\chi^2$ can achieve comparable performance with centralized correlation tests and can withstand up to 20% of clients dropping out with only minor influence on the accuracy. Besides Pearson's $\chi^2$-test, we also demonstrate how to accommodate other commonly used correlation tests such as G-test in FED-COR.

In summary, we make the following contributions:

- We propose FED-COR, the first secure federated correlation test framework. FED-COR is computation- and communication-efficient and leaks much less information than directly using secure aggregation to collect the contingency table, which completely leaks the joint distribution.
- FED-COR decomposes correlation test into frequency moments estimation that can easily be encoded/decoded using stable projection and secure aggregation techniques. We provide formal security proof and utility analysis of the protocol.
- We demonstrate how to accommodate $\chi^2$-test and G-test in FED-COR, and empirically evaluate FED-$\chi^2$ in several real-world use cases. The findings suggest that FED-$\chi^2$ can substitute centralized $\chi^2$-test with comparable accuracy. Besides, FED-$\chi^2$ can tolerate up to 20% of clients dropout with minor accuracy drop. We provide the code in the supplementary material for results verification.

## 2 RELATED WORK

There have been a line of works studying secure federated learning or statistics. Bonawitz et al. (2017) proposed the well-quoted secure aggregation protocol as a low-cost way to securely calculate linear functions in a federated setting. It has seen many variants and improvements since then. For instance, Truex et al. (2019) and Xu et al. (2019) employed advanced crypto tools for secure aggregation, such as threshold homomorphic encryption and functional encryption. So et al. (2021) proposed TURBOAGG, which combines secure sharing with erasure codes for better dropout tolerance. To improve communication efficiency, Bell et al. (2020) and Choi et al. (2020) replaced the complete graph in secure aggregation with either a sparse random graph or a low-degree graph.

Secure aggregation is deployed in a variety of applications. Agarwal et al. (2018) added binomial noise to local gradients, resulting in both differential privacy and communication efficiency. Wang et al. (2020) replaced the binomial noise with discrete Gaussian noise, which is shown to exhibit better composability. Kairouz et al. (2021) proved that the sum of discrete Gaussian is close to discrete Gaussian, thus discarding the common random seed assumption from Wang et al. (2020). The above three works all incorporate secure aggregation in their protocols to lower the noise scale required for differential privacy. Chen et al. (2020) added an extra public parameter to each client to force them to train in the same way, allowing for the detection of malicious clients during aggregation. Nevertheless, designing secure federated correlation tests, despite its importance in real-world scenarios, is not explored by existing research in this field.

## 3 METHODOLOGY

In this section, we elaborate on the design of FED-COR. Sec. 3.1 formalizes the problem, establishes the notation system, and introduces the threat model. In Sec. 3.2, we detail the design of FED-COR

by instantiating FED-COR with Pearson's $\chi^2$-test, namely FED-$\chi^2$. In Sec. 3.3 and 3.4, we present the security proof, utility analysis, communication and computation analysis of FED-$\chi^2$.

## 3.1 PROBLEM SETUP

We now formulate the problem of the federated correlation test and establish the notation system. We use $[n]$ to denote $\{1, \cdots, n\}$. We denote vectors with bold lower-case letters (*e.g.*, $\mathbf{a}, \mathbf{b}, \mathbf{c}$) and matrices with bold upper-case letters (*e.g.*, $\mathbf{A}, \mathbf{B}, \mathbf{C}$).

For the ease of representation, we use the example we mentioned in Sec. 1 to introduce all the notations. A medical company is studying the correlation between genetic defects (denoted by variable $X$) and race (denoted by variable $Y$). The support domain of $X$ (or $Y$) is denoted by $\mathcal{X}$ (or $\mathcal{Y}$). In the example, $\mathcal{X} = \{yes, no\}$ representing whether the participant has the genetic defect, and $\mathcal{Y}$ is the set of all races. We denote the size of $\mathcal{X}$ as $m_x$, the size of $\mathcal{Y}$ as $m_y$.

The company wants to use the patient records from $n$ hospitals to conduct the research. Concretely, each hospital holds a 2-dimensional local contingency table $\mathcal{D}_i = \{x \in \mathcal{X}, y \in \mathcal{Y} : v_{xy}^{(i)} \in \{0\} \cup [M]\}$, where $x$ is the row label, $y$ is the column label, and $v_{xy}^{(i)}$ is the number of patients with the label $(x, y)$. We use $m = m_x m_y$ to denote the size of the contingency table.

The first step of the traditional method in federated setting is to collect all the hospitals' contingency tables on a centralized server $\mathcal{S}$ of the company and sum them to obtain the global contingency table $\mathcal{D} = \{x, y : v_{xy} = \sum_{i \in [n]} v_{xy}^{(i)}\}$. The total number of samples with row label $x$ (or column label $y$) is defined as $v_x = \sum_{y \in \mathcal{Y}} v_{xy}$ (or $v_y = \sum_{x \in \mathcal{X}} v_{xy}$). The total number of samples observed is $v = \sum_{x \in \mathcal{X}, y \in \mathcal{Y}} v_{xy}$.

The next step is to calculate a test statistic, $s(\mathcal{D})$, on the global table. For Pearson's $\chi^2$-test, the statistic is as below:

$$s_{\chi^2}(\mathcal{D}) := \sum_{x \in \mathcal{X}, y \in \mathcal{Y}} \frac{(v_{xy} - \bar{v}_{xy})^2}{\bar{v}_{xy}}, \qquad (1)$$

where $\bar{v}_{xy} = \frac{v_x \times v_y}{v}$ is the expectation of $v_{xy}$ if $X$ and $Y$ are uncorrelated. The statistics is then compared with a threshold to decide whether $X$ and $Y$ are correlated.

Attentive readers might be aware that the method described above incurs severe ethical issues that the patient records from different hospitals are collected on a centralized server of the company, which probably violates corresponding privacy regulations. In this work, our aim is to design a secure federated correlation test protocol only leaking non-sensitive information with low computation/communication cost. Concretely, we trade off accuracy for security, as long as the estimation error is small with a high probability. Formally, if FED-COR outputs $\hat{s}$, whose corresponding standard centralized correlation test output is $s$, the following accuracy requirement should be satisfied with small multiplicative error bound $\epsilon$ and small failure probability $\delta$:

$$\mathbb{P}[(1 - \epsilon)s \leq \hat{s} \leq (1 + \epsilon)s] \geq 1 - \delta \qquad (2)$$

**Threat Model.** We assume that the centralized server $\mathcal{S}$ is honest but curious. It honestly follows the protocol due to regulatory or reputational pressure but is curious to discover extra private information from clients' legitimate updates for profit or surveillance purposes. As a result, client updates should contain as little sensitive information as feasible.

On the other hand, we assume the clients (*e.g.* the hospitals) are honest and won't collude with the server. Specifically, we do not consider client-side adversarial attacks (*e.g.*, data poisoning attacks (Bagdasaryan et al., 2020; Bhagoji et al., 2019)). However, we allow a small portion of clients to drop out during the execution. We also provide further security analysis when collusion between the server and the client happens in Appendix G.

More importantly, we assume that the marginal distributions of the variables are not sensitive while the joint distribution is. The above example is a natural case where such an assumption holds. The aggregated marginal distributions of the genetic defects and the races won't leak sensitive information. However, the correlation between a specific pair of race and genetic defect can be easily observed if the joint distribution, which is the aggregated global contingency table, is obtained by the server.

## 3.2 FEDERATED CORRELATION TEST WITH SECURE AGGREGATION

In this section, we introduce the design of FED-COR in detail by instantiating FED-COR with Pearson's $\chi^2$-test. We also discuss how the design generalizes to other statistical tests such as G-test (SOKAL et al., 1995) in Sec. 5.

**From Federated Correlation Test to Frequency Moments Estimation.** The $\alpha$-th frequency moment of a key-value stream is formally defined as below:

**Definition 1** ($\alpha$-th frequency moment). *Given a key-value stream $\{a_t \in \mathcal{A}, b_t \in \mathcal{B}\}_{t \in [T]}$, the $\alpha$-th frequency moment of $\mathcal{S}$ is defined as:*

$$\mathbb{F}_\alpha(\mathcal{S}) := \sum_{a \in \mathcal{A}} (\sum_{t \in [T]: a_t = a} b_t)^\alpha \tag{3}$$

We observe that the test statistics of many correlation tests can be rewritten as frequency moments. For example, the statistic of $\chi^2$-test can be reformatted as a second frequency moment:

$$s_{\chi^2}(\mathcal{D}) = \sum_{x,y} \frac{(v_{xy} - \bar{v}_{xy})^2}{\bar{v}_{xy}} = \sum_{x,y} (\frac{v_{xy} - \bar{v}_{xy}}{\sqrt{\bar{v}_{xy}}})^2 \tag{4}$$

In the federated setting, the $i^{th}$ client calculates the vector $\mathbf{u}_i(x, y) := \frac{v_{xy}^{(i)} - \bar{v}_{xy}/n}{\sqrt{\bar{v}_{xy}}}$, and the above formula can be rewritten as a second frequency moment estimation problem:

$$s_{\chi^2}(\mathcal{D}) = \sum_{x,y} (\frac{v_{xy} - \bar{v}_{xy}}{\sqrt{\bar{v}_{xy}}})^2 = \sum_{x,y} (\sum_{i \in [n]} \mathbf{u}_i(x, y))^2 \tag{5}$$

**Federated Frequency Moments Estimation.** Now that we have reformatted the problem, the second step is to design the messages transmitted in FED-COR for $\alpha^{th}$ frequency moments estimation. We choose *stable projection* (Indyk, 2006; Vempala, 2005) to encode the client-side information and *geometric mean estimator* (Li, 2008) to decode the aggregated message. Before we dive into the details, let's refresh some preliminaries. See Appendix A for more details on stable distribution.

**Definition 2** (Symmetric $\alpha$-stable distribution). *A random variable $X$ follows a symmetric $\alpha$-stable distribution $\mathcal{Q}_{\alpha,\beta,F}$ if its characteristic function is as follows:*

$$\phi_X(t) = \exp(-F|t|^\alpha (1 - \sqrt{-1}\beta \operatorname{sgn}(t) \tan(\frac{\pi\alpha}{2}))), \tag{6}$$

*where $F$ is the scale, $\alpha^{th} \in (0, 2]$ is the stability parameter, and $\beta$ is the skewness.*

$\alpha$-stable distribution is named due to its property called $\alpha$-stability. Briefly, the sum of independent $\alpha$-stable variables still follows an $\alpha$-stable distribution with a different scale.

**Definition 3** ($\alpha$-stability). *If random variables $X \sim \mathcal{Q}_{\alpha,\beta,1}, Y \sim \mathcal{Q}_{\alpha,\beta,1}$ and $X$ and $Y$ are independent, then $C_1 X + C_2 Y \sim \mathcal{Q}_{\alpha,\beta,C_1^\alpha + C_2^\alpha}$.*

Inspired by the idea of Indyk's well-cited paper (Indyk, 2006), we encode the frequency moments in the scale parameter of a stable distribution. To encode information contained in the local contingency table $\mathcal{D}_i$, the $i^{th}$ client collaborates with other clients to generate a projection matrix $\mathbf{P} \in \mathbb{R}^{\ell \times m}$ projection matrix, where $\ell$ is the encoding size. The components of $\mathbf{P}$ are drawn independently from an $\alpha$-stable distribution $\mathcal{Q}_{\alpha,0,1}$. The client then calculates $\mathbf{u}_i$ as defined in Eq. 5 and applies the projection get $\mathbf{e}_i := \mathbf{P} \times \mathbf{u}_i$ as the encoding (lines 1-2 in Alg. 1).

To decode, the server first sums the encodings from all the clients $\mathbf{e} := \sum_{i \in [n]} \mathbf{e}_i$. According to the $\alpha$-stability defined in Definition 3, every component $e_k$ in the encoding vector $\mathbf{e}$, $k \in [\ell]$, follows this stable distribution $\mathcal{Q}_{\alpha,0,s(\mathcal{D})}$. Thus, the statistic of the correlation test can be estimated with the scale of the distribution. We estimate the scale using an unbiased geometric mean estimator (Li, 2008) (lines 3-4 in Alg. 1).

A significant advantage of stable projection is that the encodings are linearly aggregatable and thus compatible with secure aggregation. Secure aggregation only reveals the aggregated encoding to the server and greatly reduces the privacy leakage. Furthermore, in Sec. 3.4, we show that a small encoding size suffices to accurately approximate the frequency moments with high probability and can potentially improve communication cost with certain setups.

---

**Algorithm 1** The encoding and decoding scheme (Indyk, 2006) for federated frequency moments estimation. Note that the encoding and decoding themselves do not provide any security guarantee.

---

1  **Function** ENCODE$(\boldsymbol{P}, \boldsymbol{u}_i)$:
2     |  **return** $\mathbf{P} \times \mathbf{u}_i$
3  **Function** DECODE$(\boldsymbol{e})$:
4     |  **return** $\frac{\prod_{k=1}^{\ell} |\mathbf{e}_k|^{2/\ell}}{(\frac{2}{\pi}\Gamma(\frac{2}{\ell})\Gamma(1-\frac{1}{\ell})\sin(\frac{\pi}{\ell}))^{\ell}}$               `// ℓ is the encoding size.`

---

**Algorithm 2** The complete FED-$\chi^2$ protocol. SECUREAGG is a remote procedure that receives inputs from the clients and returns the summation to the server. INITSECUREAGG is the corresponding setup protocol deciding the communication graph and other hyper-parameters.

---

1  **Round** 1: Reveal the marginal statistics
2     |  INITSECUREAGG$(n)$                            `// n: clients number`
3     |  **for** $x \in [m_x]$ **do** $v_x = $ SECUREAGG$(\{v_x^{(i)}\}_{i \in [n]})$
4     |  **for** $y \in [m_y]$ **do** $v_y = $ SECUREAGG$(\{v_y^{(i)}\}_{i \in [n]})$
5     |  **Server**
6     |     |  Calculate $v = \sum_x v_x$ and Broadcast $v$, $\{v_x\}$, and $\{v_y\}$ to all the clients
7  **Round** 2: Approximate the statistics
8     |  **Client** $i \in [n]$
9     |     |  Calculate $\bar{v}_{xy} = \frac{v_x v_y}{v}$
10    |     |  Prepare $\mathbf{u}_i$ s.t. $\mathbf{u}_i(x,y) = \frac{v_{xy}^{(i)} - \bar{v}_{xy}/n}{\sqrt{\bar{v}_{xy}}}$
11    |     |  Randomly sample a random seed $r_i$ and broadcast to all the other clients
12    |     |  Collect the random seeds from the other clients and obtain the shared random seed $r = \sum_i r_i$
13    |     |  Sample the projection matrix $\mathbf{P}$ from $\mathcal{Q}_{2,0,1}^{\ell \times m}$ using the common random seed $r$
14    |     |  Calculate $\mathbf{e}_i = $ ENCODE$(\boldsymbol{P}, \boldsymbol{u}_i)$
15    |  $\mathbf{e} = $ SECUREAGG(QUANTIZE$(\{\boldsymbol{e}_i\}_{i \in [n]})$)
16    |  **Server**
17    |     |  $\hat{s}_{\chi^2} = $ DECODE$(\boldsymbol{e})$

---

**FED-$\chi^2$ Protocol.** We instantiate FED-COR with Pearson's $\chi^2$-test, and the complete FED-$\chi^2$ protocol is presented in Alg. 2. Firstly, the marginal statistics $v_x, v_y$ and $v$ are collected with secure aggregation and broadcasted to all the clients (lines 1–6 of Alg. 2). This step can be omitted if the marginal statistics are already known. The $i^{th}$ client calculates $\mathbf{u}_i$ (lines 9–10 of Alg. 2), and samples a random seed $r_i$ and broadcasts to other clients (line 11 of Alg. 2). Then, the clients receive the random seeds and sample the projection matrix $\mathbf{P}$ from the $\alpha$-stable distribution $\mathcal{Q}_{2,0,1}^{\ell \times m}$ using the common random seed $r$ (lines 12–13 of Alg. 2). The $i^{th}$ client projects $\mathbf{u}_i$ to obtain the encoding $\mathbf{e}_i$ (line 14 of Alg. 2). Then, the encodings are quantized and aggregated with secure aggregation (line 15 of Alg. 2). As we have already known the marginal statistics in the first round, the quantization bound can be set accordingly. Additionally, we can use high precision for quantization, such as 64 bits, such that the precision of the quantized float numbers is comparable to or even better than the IEEE floating numbers. We validate this conjecture with empirical evaluation and hence ignore the effect of quantization on accuracy in the analysis. In the last step, the server gets the $\chi^2$-test statistics using the decoding algorithm described in Alg. 1 (line 17 of Alg. 2).

*Remark: Client Dropout.* Attentive readers might ask what if some clients drop out during the protocol execution? We argue that dropouts in the first round have no effect on the test's accuracy as long as the secure aggregation used is resilient to dropout, such as (Bonawitz et al., 2017; Bell et al., 2020). On the other hand, dropouts in the second round will affect the accuracy of the test. However, since the $\chi^2$ value is typically far from the decision threshold, FED-$\chi^2$ is intrinsically robust to a small portion of clients dropping out (see Section 4 for empirical assessment).

*Remark: The Selection of Secure Aggregation.* As introduced in Sec. 2, there are a variety of secure aggregation protocols for different setups (Bonawitz et al., 2017; Truex et al., 2019; Xu et al., 2019; So et al., 2021; Bell et al., 2020; Choi et al., 2020). In the rest of the paper, we choose the state-of-the-art cross-device secure aggregation protocol by Bell et al. (2020) due to its simple trust assumption and low communication cost. We want to emphasize that FED-COR can incorporate any secure aggregation protocols as needed.

### 3.3 SECURITY ANALYSIS

We now prove the security enforced by Alg. 2 via a standard simulation proof process (Lindell, 2017) on the basis of Theorem 1.

**Theorem 1** (Security). *Let $\Pi$ be an instantiation of Alg. 2 with the secure aggregation protocol in Alg. 4 of Appendix B with cryprographic security parameter $\lambda$. There exists a PPT simulator SIM such that for all clients $\mathcal{C}$, the number of clients $n$, all the marginal distributions $\boldsymbol{v}_x, \boldsymbol{v}_y$, and the aggregated encoding $\boldsymbol{e}$, the output of SIM is indistinguishable from the view of the real server $\Pi_{\mathcal{C}}$ in that execution, i.e., $\Pi_{\mathcal{C}} \approx_{\lambda} \mathrm{SIM}(\boldsymbol{e}, \boldsymbol{v}_x, \boldsymbol{v}_y, n)$.*

Intuitively, Theorem 1 illustrates that no more information about the clients except the aggregated updates is revealed to the centralized server. Note that this is the minimal necessary information for the server to estimate the test statistic. The complete proof for Theorem 1 is deferred to Appendix D.

To further emphasize the privacy protection of our protocol, we also provide analysis on the leakage when the server colludes with a client in Appendix G. We show that even the collusion happens, our protocol can still successfully hide the information in a subspace with exponential possible distributions, which practically enforce privacy given the considerably large size of the solution space.

### 3.4 UTILITY, COMMUNICATION & COMPUTATION ANALYSIS

We first present the utility analysis of FED-$\chi^2$ in Alg. 2. We show that the output of FED-$\chi^2$, $\hat{s}_{\chi^2}$, is a fairly accurate approximation (parameterized by $\epsilon$) to the correlation test output $s_{\chi^2}$ in the standard centralized setting with high probability parameterized by $\delta$ when $\ell$ is appropriately chosen. The proof is deferred to Appendix E.

**Theorem 2** (Utility). *Let $\Pi$ be an instantiation of Alg. 2 with secure aggregation protocol in Alg. 4 of Appendix B. $\Pi$ is parameterized with $\ell = \frac{c}{\epsilon^2} \log(1/\delta)$ for some constant $c$. After executing $\Pi_{\mathcal{C}}$ on all clients $\mathcal{C}$, the server yields $\hat{s}_{\chi^2}$, whose distance to the accurate correlation test output $s_{\chi^2}$ is bounded with high probability as follows:*

$$\mathbb{P}[\hat{s}_{\chi^2} < (1-\epsilon)s_{\chi^2} \lor \hat{s}_{\chi^2} > (1+\epsilon)s_{\chi^2}] \le \delta \tag{7}$$

Then we present the communication and computation cost of Alg. 2.

**Theorem 3** (Communication Cost). *Let $\Pi$ be an instantiation of Alg. 2 with secure aggregation protocol in Alg. 4 of Appendix B, then (1) the client-side communication cost is $\mathcal{O}(\log n + m_x + m_y + \ell)$; (2) the server-side communication cost is $\mathcal{O}(n \log n + nm_x + nm_y + n\ell)$.*

**Theorem 4** (Computation Cost). *Let $\Pi$ be an instantiation of Alg. 2 with secure aggregation protocol in Alg. 4 of Appendix B, then (1) the client-side computation cost is $\mathcal{O}(\log^2 n + (\ell + m_x + m_y) \log n + m\ell)$; (2) the server-side computation cost is $\mathcal{O}(n \log^2 n + n(\ell + m_x + m_y) \log n + \ell)$.*

Note that compared with the original computation cost presented in (Bell et al., 2020), the client-side overhead has an extra $\mathcal{O}(m\ell)$ term. This term is incurred by the encoding overhead. We also give an empirical evaluation on the client-side computation overhead in Sec. 4.1. Please refer to Appendix F for the detailed proof of Theorem 3 and Theorem 4.

## 4 EVALUATION

**Experiment Setup.** To assess FED-$\chi^2$'s accuracy, we simulate it on four synthetic datasets and 12 real-world datasets. We compare the multiplicative error $\hat{\epsilon} := \frac{|\hat{s}_{\chi^2}(\mathcal{D}) - s_{\chi^2}(\mathcal{D})|}{s_{\chi^2}(\mathcal{D})}$ and power of FED-$\chi^2$ with that of the standard centralized $\chi^2$-test. The four synthetic datasets are independent, linearly correlated, quadratically correlated, and logistically correlated. For the real-world datasets, we report the details in Appendix H.

We evaluate FED-$\chi^2$'s utility in two real-world application scenarios: feature selection and online false discovery rate (FDR) control. For feature selection, we report the model accuracy trained on the selected features. For online FDR control, we report the average false discovery rate. We compare the performance of FED-$\chi^2$ with that of the centralized $\chi^2$-test in each of the three experiments.

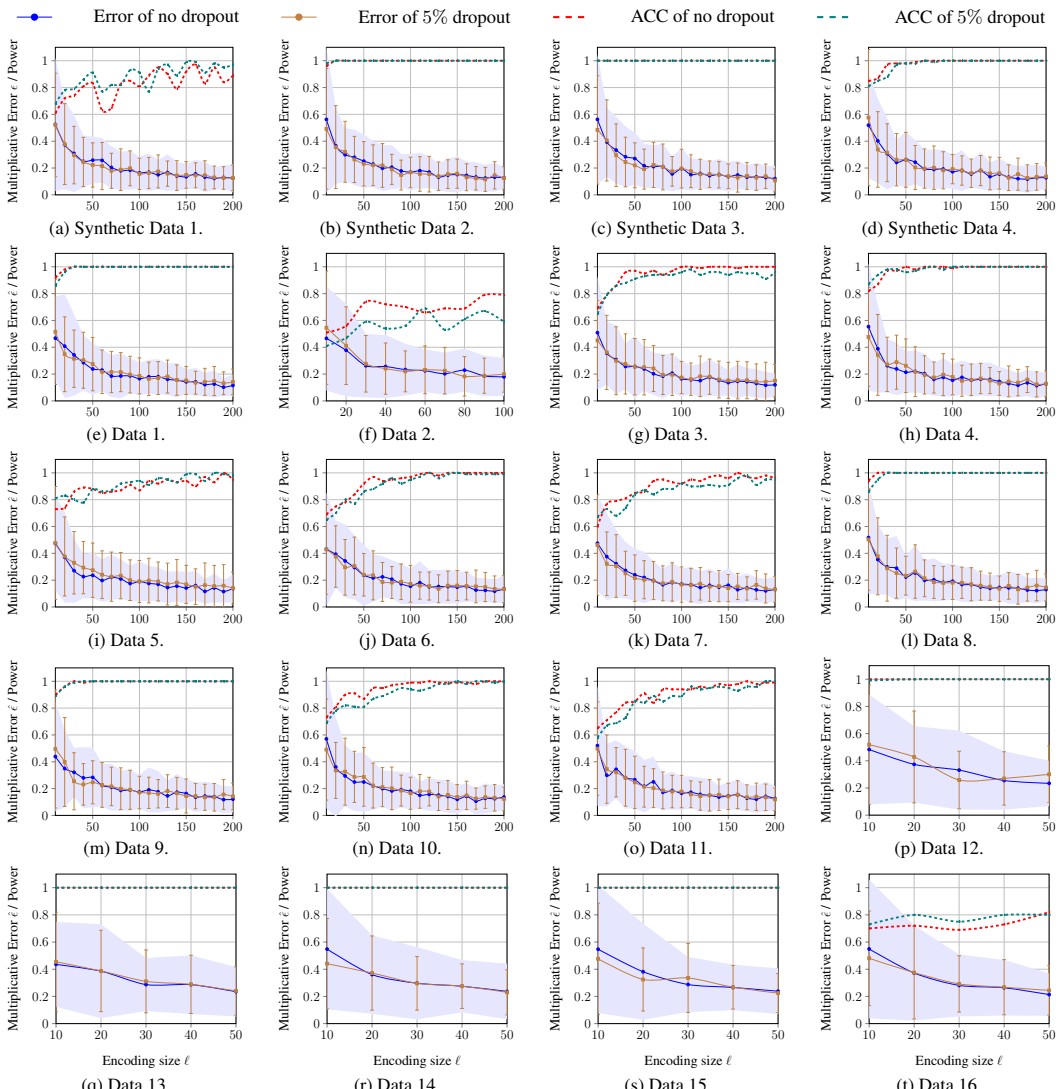

Figure 1: Multiplicative error and power of FED-$\chi^2$ w.r.t. encoding size $\ell$ with and without dropout.

For secure aggregation, we discretize all the real numbers to 64-bit fix-point numbers. We provide further evaluation on the influence of finite field size in Appendix M, which shows that FED-$\chi^2$ is numerically stable under different finite field sizes.

Unless otherwise specified, experiments are launched on an Ubuntu 18.04 LTS server equipped with 32 AMD Opteron(TM) Processor 6212 and 512GB RAM.

### 4.1 EVALUATION RESULTS

**Accuracy.** We begin by evaluating the accuracy of FED-$\chi^2$, as illustrated in Fig. 1. Each point represents the mean of 100 independent runs with 100 clients, while the error bars indicate the standard deviation. We choose $m_x = m_y = 20$ in this experiment. Note that the accuracy drop of FED-$\chi^2$ is independent of the number of clients.

From Fig. 1, we observe that the larger the encoding size $\ell$, the smaller the multiplicative error. When $\ell = 50$, the multiplicative error $\epsilon \approx 0.2$. This conforms with Theorem 2, in which the multiplicative error $\epsilon = \sqrt{\frac{c}{\ell} \log(2/\delta)}$ decreases as $\ell$ increases.

We also evaluate the power (Cohen, 2013) of FED-$\chi^2$. We set the $p$-value threshold as 0.05. From the dashed lines in Fig. 1, we can tell that the power of FED-$\chi^2$ is high. This conforms with our

observation on the multiplicative errors. Specifically, since the $\chi^2$ values are typically far from the decision threshold, a multiplicative error of $0.2$ rarely flips the final decision.

We also present the results when $5\%$ of clients drop out in the second round of FED-$\chi^2$ in Fig. 1. The results show that FED-$\chi^2$ is robust to a small portion of dropouts. In Appendix J, we present the results in terms of 10%, 15%, and 20% dropout rates. The results further show that FED-$\chi^2$ can tolerate a considerable portion of clients dropout in Round 2 of Alg. 2.

**Client-side Computation Overhead.** To assess extra computation overhead incurred by FED-$\chi^2$ on the client side, we measure the execution time of the encoding scheme on an Android 10 mobile device equipped with a Snapdragon865 CPU and 12GB RAM. We use PyDroid (Sandeep Nandal, 2020) to run the client-side computation of FED-$\chi^2$ on the Android device.

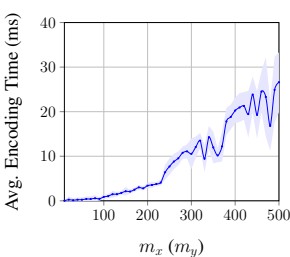

Figure 2: Client-side encoding overhead.

As shown in Fig. 2, each point represents the average of 100 separate runs, with accompanying error bars. The overhead is generally negligible. For example, for a $500 \times 500$ contingency table, the encoding takes less than 30ms. The overhead grows linearly in relation to $m_x$ ($m_y$) and consequently quadratically in Fig. 2, where $m_x = m_y$.

## 4.2 DOWNSTREAM USE CASE STUDY

**Feature Selection.** Our first case study explores secure federated feature selection using FED-$\chi^2$. The setting is that each client holds data with a large feature space and wants to collaborate with other clients to rule out unimportant features and retain features with top-$k$ highest $\chi^2$ scores. We use Reuters-21578 (Hayes & Weinstein, 1990), a standard text categorization dataset (Yang, 1999; Yang & Pedersen, 1997; Zhang & Yang, 2003), and pick the top-20 most frequent categories using 17,262 training and 4,316 test documents. These documents are distributed randomly to 100 clients, each of whom receives the same number of training documents. After removing all numbers and stop-words, we obtain 167,135 indexing terms.

The contingency table is of size $2 \times 20$ where 2 corresponds to whether a term occurs in an article and 20 corresponds to the number of different article categories. After performing feature selection using FED-$\chi^2$, we select the top 40,000 terms with the highest $\chi^2$ scores. When compared with the centralized $\chi^2$-test, 38,012 (95.03%) of the selected terms are identical, indicating that FED-$\chi^2$ produces highly consistent results with the standard $\chi^2$-test.

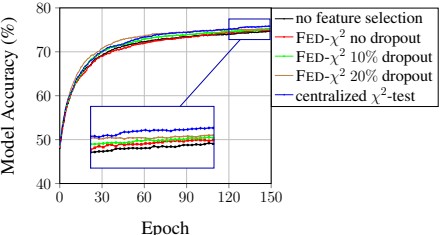

Figure 3: Accuracy of models trained with features selected by FED-$\chi^2$ and centralized $\chi^2$-test.

We then train logistic regression models using the terms selected by FED-$\chi^2$ and the centralized $\chi^2$-test, respectively. All hyper-parameters are the same. The details of these models are reported in Appendix I. The training and testing splits are the same for FED-$\chi^2$, centralized $\chi^2$-test, and model without feature selection (i.e. there are 17,262 training and 4,316 test documents). We use the same learning rate; random seed and all other settings are also the same to make the comparison fair. We get the result of Fig. 3 and the models are all trained on NVIDIA GeForce RTX 3090.

The results in Fig. 3 further demonstrate that FED-$\chi^2$ exhibits comparable performance with the centralized $\chi^2$-test. When $10\%$ and $20\%$ of clients dropout in the second round of FED-$\chi^2$, the accuracy of the trained model using the features selected by FED-$\chi^2$ does not drop much. We also examine performance without feature selection, and as expected, model accuracy is significantly greater after feature selection. Note that the model without feature selection has 2,542,700 more parameters than the model with feature selection. Hence, feature selection effectively improves model accuracy while reducing model size and computational cost. We also provide further evaluation on the influence of encoding size $\ell$ in Appendix L, which shows that FED-$\chi^2$ can achieve comparable performance with the centralized $\chi^2$-test under different $\ell$.

**Online False Discovery Rate Control.** In the third case study, we explore federated online false discovery rate (FDR) control (Foster & Stine, 2008) with FED-$\chi^2$. In an online FDR control problem,

a data analyst receives a stream of hypotheses on the database, or equivalently, a stream of $p$-values: $p_1, p_2, \cdots$. At each time $t$, the data analyst should pick a threshold $\alpha_t$ to reject the hypothesis when $p_t < \alpha_t$. The error metric is the false discovery rate, and the objective of online FDR control is to ensure that for any time $t$, the FDR up to time $t$ is smaller than a pre-determined quantity. We use the SAFFRON procedure (Ramdas et al., 2018), the state-of-the-art online FDR control, for multiple hypothesis testing. The $\chi^2$ results and corresponding $p$-values are calculated by FED-$\chi^2$. We present the SAFFRON algorithm in Appendix C.

Each time, there are 100 independent hypotheses, with a probability of 0.5 that each hypothesis is either independent or correlated. The time sequence length is 100, and the number of clients is 10. The data are synthesized from a multivariate Gaussian distribution. For the correlated data, the covariance matrix is randomly sampled from a uniform distribution. For the independent data, the covariance matrix is diagonal, and its entries are randomly sampled from a uniform distribution.

At time $t$, we use FED-$\chi^2$ to calculate the $p$-values $p_t$ of all the hypotheses, and then use the SAFFRON procedure to estimate the reject threshold $\alpha_t$ using $p_t$. The relationship between the average FDR and encoding size $\ell$ is shown in Fig. 4. We observe that the variance of independent runs is very small, so we omit the error bars. FED-$\chi^2$ achieves good performance (FDR lower than 10%) when the encoding size $l$ is larger than 200. In Fig. 4, we also provide the FDR result of the centralized $\chi^2$-test as well as the true discovery rate (TDR, i.e., #correct reject / #should reject). In addition, we provide statistics for each encoding size $l$ that was evaluated in Appendix K. The results indicate that by increasing the encoding size $\ell$, FED-$\chi^2$ can achieve comparable performance to the centralized $\chi^2$-test. The results further demonstrate that FED-$\chi^2$ can be employed in practice to facilitate online FDR control.

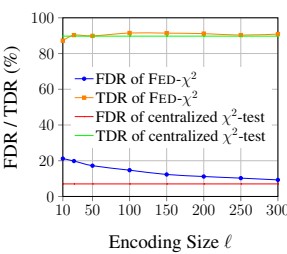

Figure 4: FDR & TDR w.r.t. $\ell$ for SAFFRON.

## 5 DISCUSSION: CORRELATION TESTS BEYOND $\chi^2$-TEST

Pearson's $\chi^2$-test is not the only correlation test compatible with FED-COR. To demonstrate the extensibility of FED-COR, we show how to recast G-test (SOKAL et al., 1995) to a frequency moments estimation problem. The reduction is more involved as the statistics in G-test contains a logarithmic term, and we rewrite $s_G$ as shown below:

$$s_G(\mathcal{D}) = 2\sum_{x,y} v_{xy} \log \frac{v_{xy}}{\bar{v}_{xy}} = 2\sum_{x,y} v_{xy} \log v_{xy} - 2\sum_{x,y} v_{xy} \log \bar{v}_{xy} \tag{8}$$

Similar to $\chi^2$-test, $\bar{v}_{xy} = \frac{v_x \times v_y}{v}$ is the expectation of $v_{xy}$ if $X$ and $Y$ are uncorrelated. The first term can be approximated using the following formula (Zhao et al., 2007) with small $\Delta$:

$$\sum_{x,y} v_{xy} \log v_{xy} = \frac{1}{2\Delta}(\sum_{x,y} v_{xy}^{1+\Delta} - \sum_{x,y} v_{xy}^{1-\Delta}) \tag{9}$$

In this way, we recast G-test to two frequency moments estimation of orders $1+\Delta$ and $1-\Delta$. The rest of the protocol is the same as FED-$\chi^2$ in Alg. 2 except that we estimate two frequency moments.

## 6 CONCLUSION & FUTURE WORKS

This paper takes an important step towards designing non-linear secure aggregation protocols in the federated setting. Specifically, we propose a universal secure protocol to evaluate frequency moments in the federated setting. We focus on an important application of the protocol: correlation test. We give formal security proof and utility analysis on our proposed protocol and validate them with empirical evaluations and downstream use case studies.

We also discuss a potential future direction. We deem it promising to provide stronger privacy guarantee for FED-COR by incorporating differential privacy techniques like differentially private frequency moments estimation (Wang et al., 2021) or adding calibrated discrete Gaussian noise (Canonne et al., 2020) to the users' local updates.

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

APPENDIX

## A   STABLE DISTRIBUTION REFRESHER

A non-degenerate distribution is said to be stable if for $X$ and $Y$ sampled from the distribution, $aX + bY$ for some constants $a, b > 0$ has the same distribution up to location and scale parameters. Paul Léby first systematically study the stable distribution family in his master piece: Calcul des probabilités (Lévy & Lévy, 1925) so stable distribution is also referred to as Léby $\alpha$-stable distribution. Stable distributions are parameterized by location $\mu$, scale $F$, the stability parameter $\alpha$ and the skewness $\beta$. When $\alpha \neq 1$, the characteristic function is as below:

$$\phi_X(t) = \exp(it\mu - F|t|^{\alpha}(1 - \sqrt{-1}\beta \operatorname{sgn}(t) \tan(\frac{\pi\alpha}{2}))) \tag{10}$$

When $\alpha = 1$, the characteristic function is given by:

$$\phi_X(t) = \exp(it\mu - F|t|^{\alpha}(1 + \frac{2}{\pi}\sqrt{-1}\beta \operatorname{sgn}(t) \log|t|)) \tag{11}$$

In the main text, we only consider a subset of stable distributions where $\mu = 0$ and $\alpha \neq 1$.

Stable distribution family contains many familiar distributions. For example, 1-stable distribution is Cauchy distribution, 2-stable distribution is Gaussian distribution, and 1/2-stable distribution is known as Lev́y distribution.

Stable distributions also have discrete analogues defined by their probability generating function

$$G(t) = exp(-Ft^{\alpha}), \tag{12}$$

where $F$ is the scale and $\alpha$ is the stability parameter. However, for discrete stable distribution, the support domain of $\alpha$ is $(0, 1]$ instead of $(0, 2]$.

## B   SECURE AGGREGATION REFRESHER

The secure aggregation protocol from Bell et al. (2020) is presented in Alg. 4. The first step of the protocol is to generate a $k$-regular graph $G$, where the $n$ vertices are the clients participating in the protocol. The server runs a randomized graph generation algorithm INITSECUREAGG presented in Alg. 3 that takes the number of clients $n$ and samples output $(G, t, k)$ from a distribution $\mathcal{D}$. In Alg. 3, we uniformly rename the nodes of a graph known as a Harary graph defined in Definition 4 with $n$ nodes and $k$ degrees. The graph $G$ is constructed by sampling $k$ neighbours uniformly and without replacement from the set of remaining $n - 1$ clients. We choose $k = \mathcal{O}(log(n))$, which is large enough to hide the updates inside the masks. $t$ is the threshold of the Shamir's Secret Sharing.

In the second step, the edges of the graph determine pairs of clients, each of which runs key agreement protocols to share random keys. The random keys will be used by each party to derive a mask for her input and enable dropouts.

In the third step, each client $c_i, i \in A_1$ sends secret share to its neighbors. In the fourth step, the server checks whether the clients dropout exceeds the threshold $\delta$, and lets the clients know their neighbors who didn't dropout.

In the fifth step, each pair $(i, j)$ of connected clients in $G$ runs a $\lambda$-secure key agreement protocol $s_{i,j} = \mathcal{KA}.Agree(sk_i^1, pk_j^1)$ which uses the key exchange in the previous step to derive a shared random key $s_{i,j}$. The pairwise masks $\mathbf{m}_{i,j} = F(s_{i,j})$ can be computed, where $F$ is the pseudorandom generator (PRG). If the semi-honest server announces dropouts and later some masked inputs of the claimed dropouts arrive, the server can recover the inputs. To prevent this happening, another level of masks, called self masks, $\mathbf{r}_i$ is added to the input. Thus, the input of client $c_i$ is: $\mathbf{y}_i = \mathbf{e}_i + \mathbf{r}_i - \sum_{j \in N_G(i), j < i} \mathbf{m}_{i,j} + \sum_{j \in N_G(i), j > i} \mathbf{m}_{i,j}$.

Steps 6–8 deal with the clients dropout by recovering the self masks $\mathbf{r}_i$ of clients who are still active and pairwise masks $\mathbf{m}_{i,j}$ of the clients who have dropped out. Finally, the server can cancel out the pairwise masks and subtract the self masks in the final sum: $\sum_{i \in A_2'} (\mathbf{y}_i - \mathbf{r}_i + \sum_{j \in NG(i) \cap (A_1' \backslash A_2'), 0 < j < i} \mathbf{m}_{i,j} - \sum_{j \in NG(i) \cap (A_1' \backslash A_2'), i < j \leq n} \mathbf{m}_{i,j})$.

**Definition 4** (HARARY$(n, k)$ Graph). *Let* HARARY$(n, k)$ *denotes a graph with $n$ nodes and degree $k$. This graph has vertices $V = [n]$ and an edge between two distinct vertices $i$ and $j$ if and only if $j - i \pmod{n} \leq (k+1)/2$ or $j - i \pmod{n} \geq n - k/2$.*

---

**Algorithm 3** INITSECUREAGG: Generate Initial Graph for SECUREAGG.

---

**Function** INITSECUREAGG($n$)**:**
 | $\triangleright$ $n$: Number of nodes.
 | $\triangleright$ $t$: Threshold of Shamir's Secret Sharing.
 | $k = \mathcal{O}(log(n))$.
 | Let $H = $ HARARY$(n, k)$.
 | Sample a random permutation $\pi : [n] \rightarrow [n]$.
 | Let $G$ be the set of edges $\{(\pi(i), \pi(j)) | (i, j) \in H\}$.
 | **return** $(G, t, k)$

---

## C SAFFRON PROCEDURE REFRESHER

In Sec. 4.2, we adopt the SAFFRON procedure (Ramdas et al., 2018) to perform online FDR control. SAFFRON procedure is currently the state of the arts for multiple hypothesis testing. In Alg. 5, we formally present the SAFFRON algorithm.

The initial error budget for SAFFRON is $(1 - \lambda_1 W_0) < (1 - \lambda_1 \alpha)$, and this will be allocated to different tests over time. The sequence $\{\lambda_j\}_{j=1}^{\infty}$ is defined by $g_t$ and $\lambda_j$ serves as a weak estimation of $\alpha_j$. $g_t$ can be any coordinate wise non-decreasing function (line 8 in Alg. 5). $R_j := I(p_j < \alpha_j)$ is the indicator for rejection, while $C_j := I(p_j < \lambda_j)$ is the indicator for candidacy. $\tau_j$ is the $j^{th}$ rejection time. For each $p_t$, if $p_t < \lambda_t$, SAFFRON adds it to the candidate set $C_t$ and sets the candidates after the $j^{th}$ rejection (lines 9-10 in Alg. 5). Further, the $\alpha_t$ is updated by several parameters like current wealth, current total rejection numbers, the current size of the candidate set, and so on (lines 11-14 in Alg. 5). Then, the decision $R_t$ is made according to the updated $\alpha_t$ (line 15 in Alg. 5).

The hyper-parameters for the SAFFRON procedure in online false discovery rate control of Sec. 4 are aligned with the setting in Ramdas et al. (2018). The target FDR level is $\alpha = 0.05$, the initial wealth is $W_0 = 0.0125$, and $\gamma_j$ is calculated in the following way: $\gamma_j = \frac{1/(j+1)^{1.6}}{\sum_{j=0}^{10000} 1/(j+1)^{1.6}}$.

## D PROOF FOR THEOREM 1

*Proof for Theorem 1.* To prove Theorem 1, we need the following lemma.

**Lemma 1** (Security of secure aggregation protocol). *Let* SECUREAGG *be the secure aggregation protocol in Alg. 4 of Appendix B instantiated with cryprographic security parameter $\lambda$. There exists a probabilistic polynomial-time (PPT) simulator* SIMSA *such that for all clients $\mathcal{C}$, the number of clients $n$, and the aggregated encoding $e$, the output of* SIMSA *is perfectly indistinguishable from the view of the real server,* i.e., SECUREAGG$_{\mathcal{C}} \approx_{\lambda}$ SIMSA$(e, n)$.

Lemma 1 is derived from the security analysis of our employed secure aggregation protocol (Theorem 3.6 in Bell et al. (2020)), which establishes that the secure aggregation protocol securely conceals the individual information in the aggregated result. With this lemma, we are able to prove the theorem for federated correlation test by presenting a sequence of hybrids that begin with real protocol execution and end with simulated protocol execution. We demonstrate that every two consecutive hybrids are indistinguishable, illustrating that the hybrids are indistinguishable according to transitivity.

HYB$_1$ This is the view of the server in the real protocol execution, REAL$_{\mathcal{C}}$.

HYB$_2$ In this hybrid, we replace the view during the execution of each SECUREAGG$(\{\mathbf{v}_x^{(i)}\}_{i \in [n]})$ in line 3 of Alg. 2 with the output of SIMSA$(\mathbf{v}_x, n)$ one by one correspondingly. According to Lemma 1, each replacement does not change the indistinguishability. Hence, HYB$_2$ is indistinguishable from HYB$_1$.

HYB$_3$ Similar to HYB$_2$, we replace the view during the execution of each SECUREAGG($\{\mathbf{v}_y^{(i)}\}_{i \in [n]}$) in line 4 of Alg. 2 with the output of SIMSA($\mathbf{v}_y, n$) one by one. According to Lemma 1, HYB$_3$ is indistinguishable from HYB$_2$.

HYB$_4$ In this hybrid, we replace the view during the execution of SECUREAGG($\{\mathbf{e}_i\}_{i \in [n]}$) in line 15 of Alg. 2 with the SIMSA($\mathbf{e}, n$). This hybrid is the output of SIM. According to Lemma 1, HYB$_4$ is indistinguishable from HYB$_3$. □

---

**Algorithm 4** SECUREAGG: Secure Aggregation Protocol. (Algorithm 2 from Bell et al. (2020))

---

**Function** SECUREAGG ($\{\boldsymbol{e}_i\}_{i \in [n]}$) :

▷ Parties: Clients $c_1, \cdots, c_n$, and Server.
▷ $l$: Vector length.
▷ $\mathbb{X}^l$: Input domain, $\mathbf{e}_i \in \mathbb{X}^l$.
▷ $F : \{0,1\}^\lambda \rightarrow \mathbb{X}^l$: PRG.
▷ *We denote by $A_1, A_2, A_3$ the sets of clients that reach certain points without dropping out. Specifically $A_1$ consists of the clients who finish step (3), $A_2$ those who finish step (5), and $A_3$ those who finish step (7). For each $A_i$, $A_i'$ is the set of clients for which the server sees they have completed that step on time.*
(1) The server runs $(G, t, k) = $ INITSECUREAGG ($n$), where $G$ is a regular degree-$k$ undirected graph with $n$ nodes. By $N_G(i)$ we denote the set of $k$ nodes adjacent to $c_i$ (its neighbors).
(2) Client $c_i, i \in [n]$, generates key pairs $(sk_i^1, pk_i^1)$, $(sk_i^2, pk_i^2)$ and sends $(pk_i^1, pk_i^2)$ to the server who forwards the message to $N_G(i)$.
(3) **for** *each Client $c_i, i \in A_1$* **do**

- Generates a random PRG seed $b_i$.
- Computes two sets of shares:

$$H_i^b = \{h_{i,1}^b, \cdots, h_{i,k}^b\} = ShamirSS(t, k, b_i)$$

$$H_i^s = \{h_{i,1}^s, \cdots, h_{i,k}^s\} = ShamirSS(t, k, sk_i^1)$$

- Sends to the server a message $m = (j, c_{i,j})$, where $c_{i,j} = \mathcal{E}_{auth}.Enc(k_{i,j}, (i||j||h_{i,j}^b||h_{i,j}^s))$, $k_{i,j} = \mathcal{KA}.Agree(sk_i^2, pk_j^2)$, for each $j \in N_G(i)$.

(4) The server aborts if $|A_1'| < (1-\delta)n$ and otherwise forwards $(j, c_{i,j})$ to client $c_j$ who deduces $A_1' \cap N_G(j)$.
(5) **for** *each Client $c_i, i \in A_2$* **do**

- Computes a shared random PRG seed $s_{i,j}$ as $s_{i,j} = \mathcal{KA}.Agree(sk_i^1, pk_j^1)$.
- Computes masks $\mathbf{m}_{i,j} = F(s_{i,j})$ and $\mathbf{r}_i = F(b_i)$.
- Sends to the server their masked input

$$\mathbf{y}_i = \mathbf{e}_i + \mathbf{r}_i - \sum_{j \in [n], j < i} \mathbf{m}_{i,j} + \sum_{j \in [n], j > i} \mathbf{m}_{i,j}$$

(6) The server collects masked inputs. It aborts if $|A_2'| < (1-\delta)n$ and otherwise sends $(A_2' \cup N_G(i), (A_1 \setminus A_2') \cup N_G(i))$ to every client $c_i, i \in A_2'$.
(7) Client $c_j, j \in A_3$ receives $(R_1, R_2)$ from the server and sends $\{(i, h_{i,j}^b)\}_{i \in R_1} \cup \{(i, h_{i,j}^s)\}_{i \in R_2}$ obtained by decrypting the $c_{i,j}$ received in Step (3).
(8) The server aborts if $|A_3'| < (1-\delta)n$ and otherwise:

- Collects, for each client $c_i, i \in A_2'$, the set $B_i$ of all shares in $H_i^b$ sent by clients in $A_3$. Then aborts if $|B_i| < t$ and otherwise recovers $b_i$ and $\mathbf{r}_i$ using the $t$ shares received which came from the lowest client IDs.
- Collects, for each client $c_i, i \in (A_1 \setminus A_2')$, the set $S_i$ of all shares in $H_i^s$ sent by clients in $A_3$. Then aborts if $|S_i| < t$ and otherwise recovers $sk_i^1$ and $\mathbf{m}_{i,j}$.
- **return** $\sum_{i \in A_2'}(\mathbf{y}_i - \mathbf{r}_i + \sum_{j \in NG(i) \cap (A_1' \setminus A_2'), 0 < j < i} \mathbf{m}_{i,j} - \sum_{j \in NG(i) \cap (A_1' \setminus A_2'), i < j \leq n} \mathbf{m}_{i,j})$.

# E    PROOF FOR UTILITY

*Proof for Theorem 2.*  First, we introduce the following lemma from Li (2008).

**Lemma 2** (Tail bounds of geometric mean estimator (Li, 2008))**.**  *The right tail bound of geometric mean estimator is:*

$$\mathbb{P}(\hat{s}_{\chi^2} - s_{\chi^2} > \epsilon s_{\chi^2}) \le \exp(-\ell \frac{\epsilon^2}{G_R}), \tag{13}$$

*where* $\frac{\epsilon^2}{G_R} = C_1 \log(1 + \epsilon) - C_1 \gamma_e(\alpha - 1) - \log(\frac{2}{\pi} \Gamma(\alpha C_1) \Gamma(1 - C_1) \sin(\frac{\pi \alpha C_1}{2}))$, $\alpha = 2$ *in our setting,* $C_1 = \frac{2}{\pi} \tan^{-1}(\frac{\log(1+\epsilon)}{(2+\alpha^2)\pi/6})$, *and* $\gamma_e$ *is the Euler's constant.*

*The left tail bound of the geometric mean estimator is:*

$$\mathbb{P}(\hat{s}_{\chi^2} - s_{\chi^2} < -\epsilon s_{\chi^2}) \le \exp(-\ell \frac{\epsilon^2}{G_L}), \tag{14}$$

*where* $\ell > \ell_0$, $\frac{\epsilon^2}{G_L} = -C_2 \log(1 - \epsilon) - \log(-\frac{2}{\pi} \Gamma(-\alpha C_2) \Gamma(1 + C_2) \sin(\frac{\pi \alpha C_2}{2})) - \ell_0 C_2 \log(\frac{2}{\pi} \Gamma(\frac{\alpha}{\ell_0}) \Gamma(1 - \frac{1}{\ell_0}) \sin(\frac{\pi}{2} \frac{\alpha}{\ell_0}))$, *and* $C_2 = \frac{12}{\pi^2} \frac{\epsilon}{(2+\alpha^2)}$.

With Lemma 2, Taking $c \ge \max(G_R, G_L)$ and $\delta = 2 \exp(-\frac{\ell \epsilon^2}{c})$, we are able to prove $\mathbb{P}[\hat{s}_{\chi^2} < (1 - \epsilon) s_{\chi^2} \vee \hat{s}_{\chi^2} > (1 + \epsilon) s_{\chi^2}] \le \delta$ with union bound, which is achieved when $\ell = \frac{c}{\epsilon^2} \log(2/\delta)$. □

# F    PROOF FOR COMMUNICATION & COMPUTATION COST

In this section, we prove Theorem 3 and Theorem 4.

**Theorem 3** (Communication Cost). Let $\Pi$ be an instantiation of Alg. 2 with secure aggregation protocol from Bell et al. (2020), then (1) the client-side communication cost is $\mathcal{O}(\log n + m_x + m_y + \ell)$; (2) the server-side communication cost $\mathcal{O}(n \log n + nm_x + nm_y + n\ell)$.

*Proof sketch for Theorem 3.*  Each client performs $k$ key agreements ($\mathcal{O}(k)$ messages, line 9 in Alg. 4) and sends 3 masked inputs ($\mathcal{O}(m_x + m_y + \ell)$ complexity, lines 3, 4, 15 in Alg. 2 and line 10 in Alg. 4). Thus, the client communication cost is $\mathcal{O}(\log n + m_x + m_y + \ell)$.

The server receives or sends $\mathcal{O}(\log n + m_x + m_y + \ell)$ messages to each client, so the server communication cost is $\mathcal{O}(n \log n + nm_x + nm_y + n\ell)$. □

---

**Algorithm 5** SAFFRON Procedure.

**Function** SAFFRONPROCEDURE ($\{p_1, p_2, \cdots\}, \alpha, W_0, \{\gamma_j\}_{j=0}^{\infty}$) **:**
  ▷ $\{p_1, p_2, \cdots\}$: Stream of $p$-values computed by FED-$\chi^2$.
  ▷ $\alpha$: Target FDR level.
  ▷ $W_0$: Initial wealth.
  ▷ $\{\gamma_j\}_{j=0}^{\infty}$: Positive non-increasing sequence summing to one.
  $i \leftarrow 0$                                         // Set rejection number.
  **for** *each $p$-value $p_t \in \{p_1, p_2, \cdots\}$* **do**
      $\lambda_t \leftarrow g_t(R_{1:t-1}, C_{1:t-1})$
      $C_t \leftarrow I(p_t < \lambda_t)$               // Set the indicator for candidacy $C_t$.
      $C_{j+} \leftarrow \sum_{i=\tau_j+1}^{t-1} C_i$    // Set the candidates after the $j^{th}$ rejection.
      **if** $t = 1$ **then**
          $\alpha_1 \leftarrow (1 - \lambda_1) \gamma_1 W_0$
      **else**
          $\alpha_t \leftarrow (1 - \lambda_t)(W_0 \gamma_{t-C_{0+}} + (\alpha - W_0) \gamma_{t-\tau_1-C_{1+}} + \sum_{j \ge 2} \alpha \gamma_{t-\tau_j-C_{j+}})$
      $R_t \leftarrow I(p_t \le \alpha_t)$                      // Output $R_t$.
      **if** $R_t = 1$ **then**
          $i \leftarrow i + 1$                  // Update rejection number.
          $\tau_i \leftarrow t$                    // Set the $i^{th}$ rejection time.
  **return** $\{R_0, R_1, \cdots\}$

**Theorem** 4 (Computation Cost). Let $\Pi$ be an instantiation of Alg. 2 with secure aggregation protocol from Bell et al. (2020), then (1) the client-side computation cost is $\mathcal{O}(m_x \log n + m_y \log n + \ell \log n + m\ell)$; (2) the server-side computation cost is $\mathcal{O}(m_x + m_y + \ell)$.

*Proof sketch for Theorem 4.* Each client computation can be broken up as $k$ key agreements ($\mathcal{O}(k)$ complexity, line 9 in Alg. 4), generating masks $\mathbf{m}_{i,j}$ for all neighbors $c_j$ ($\mathcal{O}(k(m_x + m_y + \ell))$ complexity, lines 3, 4, 15 in Alg. 2 and line 10 in Alg. 4), sampling encoding matrix $\mathbf{P}$ cost $\mathcal{O}(m\ell)$, line 13 in Alg. 2, and encoding computation cost $\mathcal{O}(m\ell)$ (line 14 in Alg. 2). Thus, the client computation cost is $\mathcal{O}(m_x \log n + m_y \log n + \ell \log n + m\ell)$.

The server-side follows directly from the semi-honest computation analysis in Bell et al. (2020). The extra $\mathcal{O}(\ell)$ term is the complexity of the geometric mean estimator.

$\square$

## G  FURTHER SECURITY ANALYSIS WHEN COLLUSION HAPPENS

We have shown that Alg. 2 provides strong security guarantee when there is no collusion between the clients and the server. That is, the server only knows the non-private marginal distribution of the contingency table and the final aggregated results. In the following section, we will analyze the leakage of Alg. 2 when the collusion happens to demonstrate that FED-$\chi^2$ provides strong privacy guarantee and also help the readers better understand our protocol.

*Remark: what does Alg. 2 leak when collusion between the server and the client happens?* If the server colludes with one client, then it knows the random seed $r$ (line 12 of Alg. 2) used to generate the projection matrix $\mathbf{P}$. In the following, we will analyze the leakage of client private data when the server knows $\mathbf{P}$.

By Theorem 1, we show that individual updates of clients are perfectly hidden in the aggregated results and FED-$\chi^2$ leaks no more than a linear equation system if the server knows $\mathbf{P}$:

$$\begin{cases} \mathbf{P} \times \mathbf{v} &= \mathbf{e}^T \\ \mathbf{J}_{1,m_y} \times \mathbf{V}^T &= \mathbf{v}_x^T \\ \mathbf{J}_{1,m_x} \times \mathbf{V} &= \mathbf{v}_y^T \end{cases}, \tag{15}$$

where $\mathbf{J}_{1,m_x}$ and $\mathbf{J}_{1,m_y}$ are $1 \times m_x$ and $1 \times m_y$ unit matrices, $\mathbf{V}$ is an $m_x \times m_y$ matrix whose elements are $\{v_{xy}\}$, and $\mathbf{v}$ is the flattened vector of $\mathbf{V}$.

To understand (15), $\mathbf{v}$ (or $\mathbf{V}$) is sensitive and all the other matrices and vectors are already known to the server. Also note that due to the requirement of secure aggregation, all the values in (15) are discretized into a finite field. Thus, the server can solve the system of equations (15) on a finite field to get information about $\mathbf{v}$. The following theorem establishes an important fact: the above equation system has a large solution space, which conceals the real joint distribution.

**Proposition 1.** *Given a projection matrix $\boldsymbol{P} \in \mathbb{Z}_q^{\ell \times m}$, $\boldsymbol{v}_x \in \mathbb{Z}_q^{m_x}$, $\boldsymbol{v}_y \in \mathbb{Z}_q^{m_y}$ and $\boldsymbol{e} \in \mathbb{Z}_q^{\ell}$, if $m > \ell + m_x + m_y$, there are at least $q^{m-\ell-m_x-m_y}$ solutions to the system of equations (15).*

*Proof sketch for Proposition 1.* The system of linear equations on $\mathbb{Z}_q$ contains $m_x + m_y + \ell$ equations and $m$ variables. Given $m > m_x + m_y + \ell$, the rank of the coefficient matrix is no more than $m_x + m_y + \ell$. According to the Rouché–Capelli theorem (Brunetti & Renato, 2014) on finite fields, the solution forms a at least $m - m_x - m_y - \ell$-dimensional traslation of subspace of $\mathbb{Z}_q^m$. As a result, we know that the solution space contains at least $q^{m-\ell-m_x-m_y}$ solution vectors. $\square$

Theorem 1 shows an important fact that the joint distribution is hidden in a subspace with exponential possible distributions. Although the collusion between the client and the server is not likely to happen in the cross-silo federated settings (consider our example in Sec. 3.1) and thus not considered in our threat model, we still show that Alg. 2 practically enforce privacy given the considerably large size of the solution space.

## H  DETAILS OF DATASETS

The details for the real-world datasets used in Sec. 4.1 are provided in Table 1. The license of Credit Risk Classification (Govindaraj, Praveen) is CC BY-SA 4.0, the license of German Traffic Sign (Houben et al., 2013) is CC0: Public Domain. Other datasets without a license are from UCI Machine Learning Repository (Dua & Graff, 2017).

Table 1: Dataset details.

| ID | Data | Attr #1 | A#1 Cat | Attr #2 | A#2 Cat |
|---|---|---|---|---|---|
| 1 | Adult Income (Kohavi, 1996; Kohavi, Ronny and Becker, Barry) | Occupation | 14 | Native Country | 41 |
| 2 | Credit Risk Classification (Govindaraj, Praveen) | Feature 6 | 14 | Feature 7 | 11 |
| 3 | Credit Risk Classification (Govindaraj, Praveen) | Credit Product Type | 28 | Overdue Type I | 35 |
| 4 | Credit Risk Classification (Govindaraj, Praveen) | Credit Product Type | 28 | Overdue Type II | 35 |
| 5 | Credit Risk Classification (Govindaraj, Praveen) | Credit Product Type | 28 | Overdue Type III | 36 |
| 6 | German Traffic Sign (Houben et al., 2013) | Image Width | 219 | Traffic Sign | 43 |
| 7 | German Traffic Sign (Houben et al., 2013) | Image Height | 201 | Traffic Sign | 43 |
| 8 | German Traffic Sign (Houben et al., 2013) | Upper left X coordinate | 21 | Traffic Sign | 43 |
| 9 | German Traffic Sign (Houben et al., 2013) | Upper left Y coordinate | 16 | Traffic Sign | 43 |
| 10 | German Traffic Sign (Houben et al., 2013) | Lower right X coordinate | 204 | Traffic Sign | 43 |
| 11 | German Traffic Sign (Houben et al., 2013) | Lower right Y coordinate | 186 | Traffic Sign | 43 |
| 12 | Mushroom (Schlimmer, Jeff) | Cap color | 10 | Odor | 9 |
| 13 | Mushroom (Schlimmer, Jeff) | Gill color | 12 | Stalk color above ring | 9 |
| 14 | Mushroom (Schlimmer, Jeff) | Stalk color below ring | 9 | Ring Type | 8 |
| 15 | Mushroom (Schlimmer, Jeff) | Spore print color | 9 | Habitat | 7 |
| 16 | Lymphography (Kononenko, Igor and Cestnik, Bojan) | Structure Change | 8 | No. of nodes | 8 |

## I  DETAILS OF REGRESSION MODELS

The details of the regression models trained in feature selection in Sec. 4.2 is reported in Table 2. The training and testing splits are the same for FED-$\chi^2$, centralized $\chi^2$-test and model without feature selection (i.e. there are 17,262 training and 4,316 test documents). We use the same learning rate; random seed and all other settings are also the same to make the comparison fair. We get the result of Fig. 3 and the models are all trained on NVIDIA GeForce RTX 3090.

## J  FURTHER RESULTS ON FED-$\chi^2$ WITH DROPOUTS

We present the results of 10%, 15%, and 20% clients dropout in Fig. 5. The results further show that FED-$\chi^2$ can tolerate a considerable portion of clients dropout in Round 2 of Alg. 2.

## K  FURTHER RESULTS FOR ONLINE FDR CONTROL

In this section, we provide further results for online FDR control. As we have shown in Fig. 4, FED-$\chi^2$ achieves good performance when the encoding size $l$ is larger than 200. In addition, we provide statistics for each encoding size $l$ that was evaluated in Table 3. These results demonstrate that FED-$\chi^2$ performs well and is comparable to the centralized $\chi^2$-test when the encoding size $l$ is increased.

## L  FURTHER RESULTS FOR FEATURE SELECTION

Our results in Sec. 4.2, paragraph **Feature Selection**, demonstrate that FED-$\chi^2$ performs well when encoding size $l = 50$. We conduct experiments with different encoding sizes $l$ to further assess their effect on FED-$\chi^2$'s performance. In Fig. 6, we present the effect of encoding size $l$ on the ratio of the commonly-selected features between the original centralized $\chi^2$-test and FED-$\chi^2$. A larger ratio of commonly-selected features means that FED-$\chi^2$ performs more closely to the original centralized $\chi^2$-test. And if the ratio is 1, these two algorithms select the identical features. The results in Fig. 6 show that when the encoding size $l$ increases, the performance of FED-$\chi^2$ approaches that of the original centralized $\chi^2$-test.

Similar to Sec. 4.2, we evaluate FED-$\chi^2$'s performance under different encoding sizes $l$ by training the model with the features selected by FED-$\chi^2$. Fig. 7 shows the results. When trained with FED-$\chi^2$-selected features, the model can achieve comparable accuracy to the model trained with features

Table 2: Model details.

| Task | Model Size | Learning Rate | Random Seed |
|---|---|---|---|
| FED-$\chi^2$ | $40000 \times 20$ | 0.1 | 0 |
| Centralized $\chi^2$-test | $40000 \times 20$ | 0.1 | 0 |
| Without Feature Selection | $167135 \times 20$ | 0.1 | 0 |

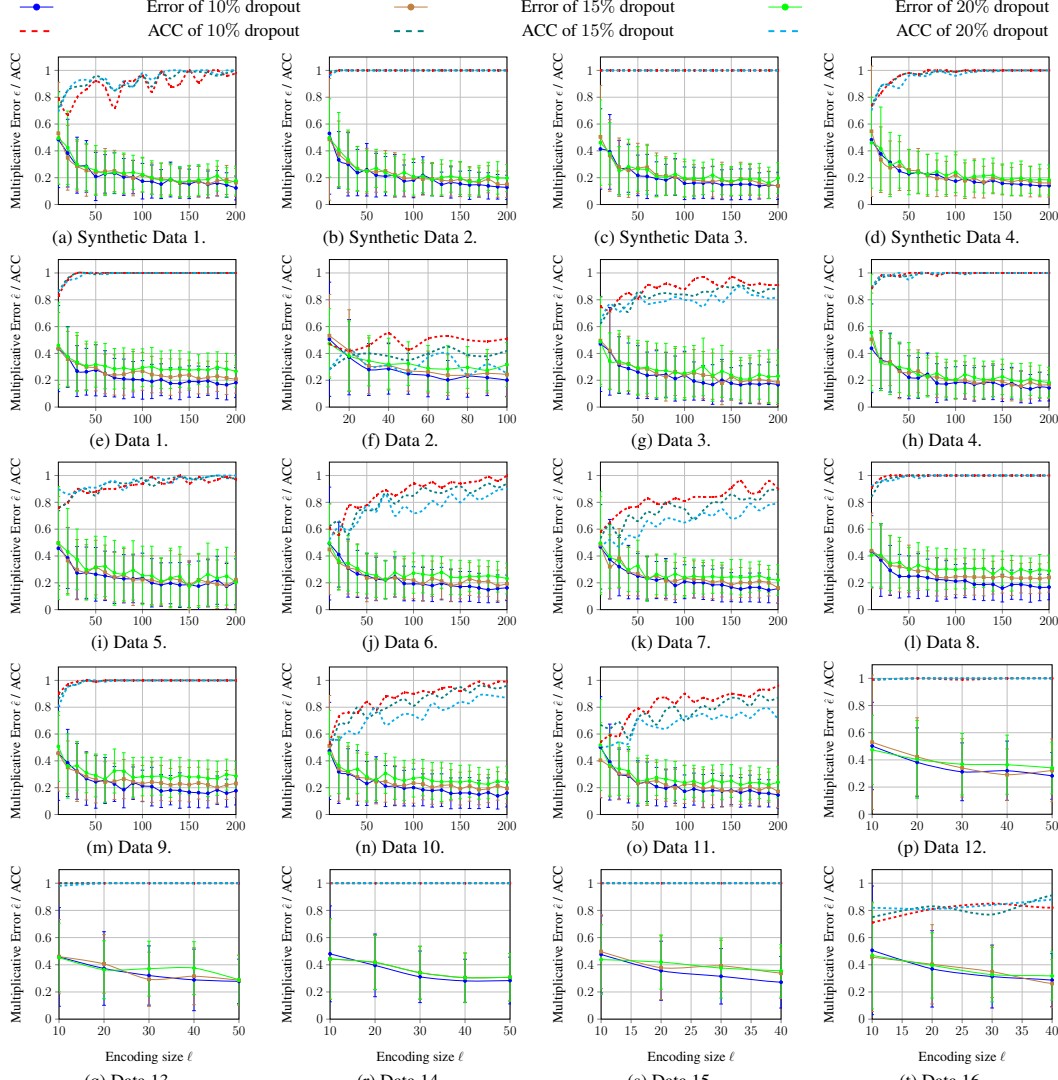

Figure 5: Multiplicative error and accuracy of FED-$\chi^2$ w.r.t. encoding size $\ell$ w/ and w/o dropout.

selected by the original centralized $\chi^2$-test. Also, consistent with the results in Fig. 3 in Sec. 4.2, we see that when the encoding size $l \geq 25$, models trained by FED-$\chi^2$-selected features achieve higher accuracy than that of the models without feature selection. These results further demonstrate the effectiveness of FED-$\chi^2$.

# M    INFLUENCE OF FINITE FIELD SIZE

As shown in Fig. 8, we test the performance of FED-$\chi^2$ under different finite field size $q$. We observe that when $q \in \{2^{16}, 2^{32}, 2^{64}\}$, there is almost no difference in the performance. The result shows that FED-$\chi^2$ is numerically stable.

Table 3: Detailed results for online FDR control.

| | #should reject | #should accept | #correct reject | #false reject |
|---|---|---|---|---|
| FED-$\chi^2$, $l = 10$ | 5,900 | 4,100 | 5,144 | 1,392 |
| FED-$\chi^2$, $l = 25$ | 5,900 | 4,100 | 5,328 | 1,356 |
| FED-$\chi^2$, $l = 50$ | 5,900 | 4,100 | 5,325 | 1,143 |
| FED-$\chi^2$, $l = 100$ | 5,900 | 4,100 | 5,398 | 943 |
| FED-$\chi^2$, $l = 150$ | 5,900 | 4,100 | 5,393 | 765 |
| FED-$\chi^2$, $l = 200$ | 5,900 | 4,100 | 5,377 | 687 |
| FED-$\chi^2$, $l = 250$ | 5,900 | 4,100 | 5,328 | 615 |
| FED-$\chi^2$, $l = 300$ | 5,900 | 4,100 | 5,361 | 556 |
| centralized $\chi^2$-test | 5,900 | 4,100 | 5,294 | 408 |

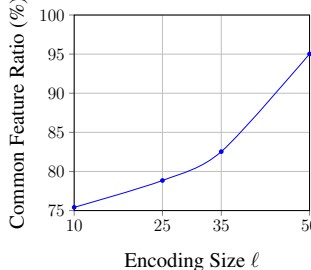

Figure 6: Ratio of commonly-selected features between FED-$\chi^2$ and original centralized $\chi^2$-test.

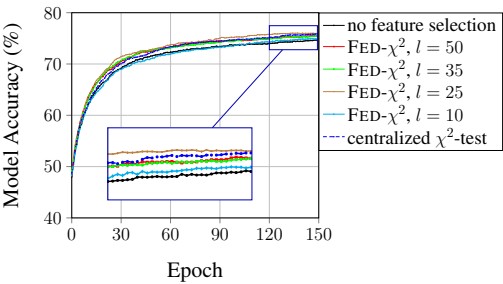

Figure 7: Accuracy of model trained w/ FED-$\chi^2$-select features under different encoding size $l$.

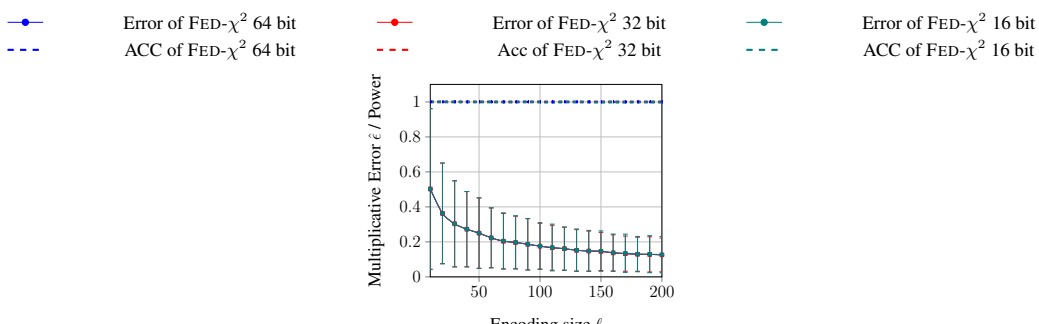

Figure 8: Performance of FED-$\chi^2$ with different finite field size on synthetic data 3.

