# OpenReview forum: "Fed-Cor: Federated Correlation Test with Secure Aggregation"
_ICLR.cc/2023/Conference — Submitted to ICLR 2023_

### Official Review · Reviewer_jFiZ · 2022-10-28

**Confidence:** 4
**Correctness:** 4
**Technical Novelty And Significance:** 2
**Empirical Novelty And Significance:** 4
**Recommendation:** 3

**Clarity, Quality, Novelty And Reproducibility:**

The writing is very clear.
The novelty is weak since most of the work is done by stable projections.

**Strength And Weaknesses:**

Strengths:
1. The writing is very clear.
2. The problem is well motivated and the explanation of the model using an example makes the paper easy to understand.
3. The experiments seem thorough and the evaluation is done on both real world and synthetic datasets.

Weaknesses:

The idea of correlation can be calculated using moments is quite obvious and if one knows the random projections literature, the idea of using stable projections to calculate $p^{th}$ moments for $p \leq 2$ is also standard. The paper claims to make headway into non-linear secure aggregation, but using random (linear) projections in secure aggregations is not new and extending it to stable projections is great, but not enough of a contribution in and of itself, in my opinion.

The experiments are extensive and thorough, but besides the experiments studying the effects of clients dropping out, the experiments provide little additional info since the empirical results are expected from the theoretical results and they just illustrate the accuracy of stable random projections.

Things that did not impact my score:

The formal security proof and utility proofs are just corollaries of the results from previous papers, and could be stated as such instead of "theorems". The communication and computation costs can just be remarks since they are simple calculations.


**Summary Of The Paper:**

This paper considers the problem of correlation testing in the federated framework compatible with secure aggregation. They motivate the problem of correlation testing with several applications and also give a relevant hospitals based example for the federated setting. They propose an algorithm with two simple observations - calculating correlations is equivalent to calculating moments, and one can use stable random projections to calculate moments. They provide formal security guarantees, utility guarantees and calculations of communication and computation costs. Lastly, they conduct extensive experiments on real world and simulated datasets to illustrate the efficacy of their algorithm.

**Summary Of The Review:**

I think the weaknesses of the paper outweigh the strengths and the contribution is not novel enough to be published at ICLR.

I really liked reading the paper since it taught me something and instructed me to add the idea of stable random projections to my toolbox. I think this paper should definitely be presented at workshops to spread the idea and submitting it to a short journal like Statistics and Probability Letters or some other empirical statistics journals which accepts short paper contributions.

---

### Official Review · Reviewer_Nyg7 · 2022-10-29

**Confidence:** 3
**Correctness:** 4
**Technical Novelty And Significance:** 2
**Empirical Novelty And Significance:** 2
**Recommendation:** 6

**Clarity, Quality, Novelty And Reproducibility:**

Novelty is the least impressive part of paper. Authors had access to 2 separate methods and described a solution com-positioning them. While this is not a grounds for rejection - it weakens the contributions. Especially when both methods are not described in any detail.

Paper will be hard to reproduce as methods used are not described but rather stub functions are used

The math behind the paper is not clear! The math mentioned in the paper captures simple concepts around breaking down of correlation tests into moments. The actual math entailing projection and then aggregation is missing.

**Strength And Weaknesses:**

S1. Paper proposes a method for doing federated correlation in a distributed setting

Weaknesses:
1. The paper relies heavily on 2 separate works (one of which is quite recent). Work 1- is by Indyk et. al. for federated frequency moments estimation and Work 2 - is Bell et. al. Secure aggregation of frequency moments.  These two methods make the paper function as in paper can be summarized as composition of the two papers. Work 2 - can be replaced by any other scheme for secure aggregation. Both of these are left for readers to learn on their own! A novice reader could feel lost as math for simple stuff (like breaking of chi square into moments) is provided but the actual method (both federated computation of frequency moments) and secure aggregation are taken for granted. It would be greatly beneficial to community if authors could take the time (and pages left) to actually describe the methods in simple terms.
2. Computations analysis seems to exclude the time required in actual secure aggregation.
3. Paper lacks reproducibility statement


**Summary Of The Paper:**

The paper proposed a federated framework for generating correlation tests for data distributed over multiple clients. The clients collaboratively generate a shared projection matrix, then do a projection and secure aggregation is applied to prevent data leakage from individual clients.

**Summary Of The Review:**

Based on the work done in the paper, since there is missing math and explanation and the paper has weak experimentation and limited novelty its hard to accept. Additionally paper may be mis classified as a machine learning paper - I think it belongs in a stats journal.

---

### Official Review · Reviewer_2ZL2 · 2022-10-31

**Confidence:** 3
**Clarity, Quality, Novelty And Reproducibility:** As mentioned before, the write-up is …
**Correctness:** 3
**Technical Novelty And Significance:** 4
**Empirical Novelty And Significance:** 2
**Recommendation:** 6

**Strength And Weaknesses:**

Strength:
- The problem considered is important.
- The proposed solution is intuitive and not too complex. It's also backed by theoretical analysis and security proof.
- Numerical experiments are comprehensive and somewhat convincing.

Weakness:
- This write-up seems to be very tied to the chi-square test.
- Is it the case that, in most cases, the weight updates already reveal a lot of information? Then lack of differential privacy could be very critical.

**Summary Of The Paper:**

This paper proposed a federated framework for correlation tests. And demonstrates an empirical evaluation of chi-square test and G-test under this framework.

**Summary Of The Review:**

Overall I think it's a good paper, may need a better write-up.

---

### Decision · Program_Chairs · 2023-01-20

**Decision:**

Reject

**Justification For Why Not Higher Score:**

No author response was provided.

**Justification For Why Not Lower Score:**

No author response was provided.

**Metareview: Summary, Strengths And Weaknesses:**

The reviewers had some major concerns which were not alleviated since no author response was provided. Hopefully, in the revised version of the paper the concerns of the reviewers will be addressed.